# Diversity and evolution of the emerging *Pandoraviridae* family

Matthieu Legendre [1], Elisabeth Fabre[1], Olivier Poirot[1], Sandra Jeudy[1], Audrey Lartigue[1], Jean-Marie Alempic[1], Laure Beucher[2], Nadège Philippe [1], Lionel Bertaux[1], Eugène Christo-Foroux[1], Karine Labadie[3], Yohann Couté [2], Chantal Abergel [1] & Jean-Michel Claverie[1]

With DNA genomes reaching 2.5 Mb packed in particles of bacterium-like shape and dimension, the first two Acanthamoeba-infecting pandoraviruses remained up to now the most complex viruses since their discovery in 2013. Our isolation of three new strains from distant locations and environments is now used to perform the first comparative genomics analysis of the emerging worldwide-distributed *Pandoraviridae* family. Thorough annotation of the genomes combining transcriptomic, proteomic, and bioinformatic analyses reveals many non-coding transcripts and significantly reduces the former set of predicted protein-coding genes. Here we show that the pandoraviruses exhibit an open pan-genome, the enormous size of which is not adequately explained by gene duplications or horizontal transfers. As most of the strain-specific genes have no extant homolog and exhibit statistical features comparable to intergenic regions, we suggest that de novo gene creation could contribute to the evolution of the giant pandoravirus genomes.

[1] Aix Marseille Univ, CNRS, Structural and Genomic Information Laboratory, UMR 7256 (IMM FR 3479), 163 Avenue de Luminy, Case 934, 13288 Marseille cedex 9, France. [2] Univ. Grenoble Alpes, CEA, Inserm, BIG-BGE, 38000 Grenoble, France. [3] CEA-Institut de Génomique, GENOSCOPE, Centre National de Séquençage, 2 rue Gaston Crémieux, CP5706, 91057 Evry Cedex, France. Correspondence and requests for materials should be addressed to M.L. (email: legendre@igs.cnrs-mrs.fr) or to C.A. (email: Chantal.Abergel@igs.cnrs-mrs.fr) or to J.-M.C. (email: Jean-Michel.Claverie@univ-amu.fr)

For 10 years after the serendipitous discovery of the first giant virus (i.e., easily visible by light microscopy) Acanthamoeba polyphaga Mimivirus[1, 2], environmental sampling in search of other Acanthamoeba-infecting viruses only succeeded in the isolation of additional members of the *Mimiviridae* family[3, 4]. Then, when we returned in 2013 to the Chilean coastal area from where we previously isolated *Megavirus chilensis*[3], we isolated the even bigger *Pandoravirus salinus*[5]. Its unique characteristics suggested the existence of a different family of giant viruses infecting Acanthamoeba. The worldwide distribution of this predicted virus family, the proposed *Pandoraviridae*, was quickly hinted by our subsequent isolation of *Pandoravirus dulcis* more than 15 000 km away, in a freshwater pond near Melbourne, Australia[5]. We also spotted pandoravirus-like particles in an article reporting micrographs of Acanthamoeba infected by an unidentified "endosymbiont"[6], the genome sequence of which has recently become available as that of the German isolate *Pandoravirus inopinatum*[7].

Here we describe three new members of the proposed *Pandoraviridae* family that were isolated from different environments and distant locations: *Pandoravirus quercus*, isolated from ground soil in Marseille (France); *Pandoravirus neocaledonia*, isolated from the brackish water of a mangrove near Noumea airport (New Caledonia); and *Pandoravirus macleodensis*, isolated from a freshwater pond near Melbourne (Australia), only 700 m away from where we previously isolated *P. dulcis*. Following the characterization of their replication cycles in *Acanthamoeba castellanii* by light and electron microscopy, we analyzed the five pandoravirus strains available in our laboratory through combined genomic, transcriptomic, and proteomic approaches. We then used these data (together with the genome sequence of *P. inopinatum*) in a comparative manner to build a global picture of the emerging family and refine the genome annotation of each individual strain. While the number of encoded proteins has been revised downward, we unraveled hundreds of previously unpredicted genes associated to non-coding transcripts. From the comparison of the six representatives at our disposal, the *Pandoraviridae* family appears quite diverse in terms of gene content, consistent with a family for which many members are still to be isolated. A large fraction of the pan-genome codes for proteins without homologs in cells or other viruses, raising the question of their origin. The purified virions are made of more than 200 different proteins, about half of which are shared by all tested strains in well-correlated relative abundances. This large core proteome is consistent with the highly similar early infection stages exhibited by the different isolates.

## Results

**Environmental sampling and isolation of pandoravirus strains**. We used the same isolation protocol that led to the discovery of *P. salinus* and *P. dulcis*[5]. It consists in mixing the sampled material with cultures of Acanthamoeba adapted to antibiotic concentrations high enough to inhibit the growth of other environmental microorganisms (especially bacteria and fungi). Samples were taken randomly from humid environments susceptible to harbor Acanthamoeba cells. This led to the isolation of three new pandoravirus strains: *P. quercus*; *P. neocaledonia*; and *P. macleodensis* (Table 1, see Methods). They exhibit adequate divergence to start assessing the conserved features and the variability of the emerging *Pandoraviridae* family. When appropriate, our analyses also include data from *P. inopinatum*, isolated in a German laboratory from a patient with an Acanthamoeba keratitis[7].

**Study of the replication cycles and virion ultrastructures**. Starting from purified particles inoculated into *A. castellanii* cultures, we analyzed the infectious cycle of each isolate using both light and transmission electron microscopy (ultrathin section). As previously observed for *P. salinus* and *P. dulcis*, the replication cycles of these new pandoraviruses were found to last an average of 12 h[5] (8 h for the fastest *P. neocaledonia*). The infectious process is the same for all viruses, beginning with the internalization of individual particles by Acanthamoeba cells. Following the opening of their apical pore, the particles ("pandovirions") transfer their translucent content to the cytoplasm through the fusion of the virion internal membrane with that of the phagosome. The early stage of the infection is remarkably similar for all isolates. While we previously reported that the cell nucleus was fully disrupted during the late stage of the infectious cycle[5], the thorough observation of the new strains revealed neo-synthetized particles in the cytoplasm of cells still exhibiting nucleus-like compartments in which the nucleolus was no longer recognizable (Supplementary Fig. 1). Eight hours post infection, mature virions became visible in vacuoles and are released through exocytosis (Supplementary Movie). For all isolates, the

**Table 1 Data on the pandoravirus isolates used in this work**

| Name | Origin | Isolate | RNA-seq | Virion proteome | Genome size (bp) (G + C)% | N ORFs[a] (standard) | N Genes (stringent) |
|---|---|---|---|---|---|---|---|
| *P. salinus* | Chile | Ref. [5] | This work | This work | 2473870 62% | 2394 (2541)[a] | 1430 ORFs 214 lncRNAs 3 tRNAs |
| *P. dulcis* | Australia | Ref. [5] | This work | This work | 1908524 64% | 1428 (1487)[a] | 1070 ORFs 268 lncRNAs 1 tRNA |
| *P. quercus* | France (Marseille) | This work | This work | This work | 2077288 61% | 1863 | 1185 ORFs 157 lncRNAs 1 tRNA |
| *P. neocaledonia* | New Caledonia | This work | This work | This work | 2003191 61% | 1834 | 1081 ORFs 249 lncRNAs 3 tRNA |
| *P. macleodensis* | Australia (Melbourne) | This work | — | — | 1838258 58% | 1552 | 926 ORFs 1 tRNA |
| *P. inopinatum* | Germany | Ref. [6] | — | — | 2243109 61% | 2397 (1839)[a] | 1307 ORFs 1 tRNA |

*NC* non-protein-coding transcripts
[a]The number of ORFs predicted in the original publications are indicated in parenthesis below the number obtained using the same standard reannotation protocol for all genomes. A more stringent estimate (next column) is taking into account protein sequence similarity as well as RNA-seq and proteomic data when available (see Methods)

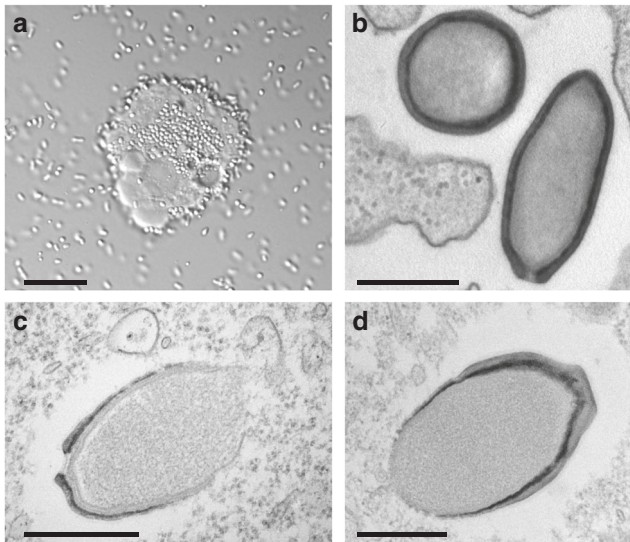

**Fig. 1** The new pandoravirus isolates. **a** Overproduction by an *A. castellanii* cell of *Pandoravirus macleodensis* virions from the environmental sample prior cell lysis. Environmental bacteria can be seen in the culture medium together with *P. macleodensis* virions. (scale bar is 10 μm). **b** TEM image of an ultrathin section of *A. castellanii* cell during the early phase of infection by *P. neocaledonia*. The ameba pseudopods are ready to engulf the surrounding virions. Ten minutes pi, virions have been engulfed and are in vacuoles (scale bar is 500 nm). **c** TEM image of an ultrathin section of *A. castellanii* cell during the assembly process of a *P. salinus* virion (scale bar is 500 nm). **d** TEM image of an ultrathin section of a nascent *P. quercus* virion. (scale bar is 500 nm). The structures of the mature particles from the different strains do not exhibit any noticeable difference

replicative cycle ends with the cells lysis and the release of about a hundred particles (Fig. 1).

**Genome sequencing and annotation**. Genomic DNA of *P. neocaledonia*, *P. macleodensis*, and *P. quercus* were prepared from purified particles and sequenced using either the PacBio or Illumina platforms (see Methods). As for *P. salinus*, *P. dulcis*[5], and *P. inopinatum*[7], the three new genomes assembled as single linear double-stranded DNA (dsDNA) molecules (≈60% G + C) with sizes ranging from 1.84 to 2 Mb. In addition to their translucent amphora-shaped particles (Fig. 1), higher than average G + C content and genomic gigantism thus remain characteristic features shared by the *Pandoraviridae*[5, 8]. Given the high proportion of viral genes encoding proteins without database homolog, gene predictions based on purely ab initio computational approaches (i.e., "ORFing" and coding propensity estimates) are notoriously unreliable, leading to inconsistencies between teams using different values of arbitrary parameters (e.g., minimal open reading frame (ORF) size). For instance among families of large dsDNA viruses infecting eukaryotes, the average protein-coding gene density reportedly varies from one gene every 335 bp (*Phycodnaviridae*, NCBI: NC_008724) up to one gene every 2120 bp (*Herpesviridae*, NCBI: NC_003038), while the consensus is clearly around one gene every kb (such as for bacteria). As a result, one oscillates between situations where many genes are overpredicted and others where many real genes are probably overlooked. Such uncertainty about which genes are "real" introduces a significant noise in comparative genomic analyses and the subsequent testing of evolutionary hypotheses. In addition, computational methods are mostly blind to genes expressed as non-protein-coding transcripts.

To overcome the above limitations, we performed strand-specific RNA-seq experiments and particle proteome analyses, the results of which were mapped on the genome sequences. Only genes supported by experimental evidence (or protein similarity) were retained into this stringent reannotation protocol (see Methods, Supplementary Fig. 2). On one hand, this new procedure led to a reduced set of predicted proteins, on the other hand it allowed the discovery of an unexpected large number of non-coding transcripts (Table 1).

The new set of validated protein-coding genes exhibits a strongly diminished proportion of ORFs shorter than 100 residues, most of which are unique to each pandoravirus strain (Supplementary Fig. 3). The stringent annotation procedure also resulted in genes exhibiting a well-centered unimodal distribution of codon adaptation index (CAI) values (Supplementary Fig. 3).

For consistency, we extrapolated our stringent annotation protocol to *P. inopinatum* and *P. macleodensis*, reducing the number of predicted proteins taken into account in further comparisons (see Methods, Table 1). As expected, the discrepancies between the standard versus stringent gene predictions are merely due to the overprediction of small ORFs (length < 300 nucleotides). Such arbitrary ORFs are prone to arise randomly in G + C-rich sequences within which stop codons (TAA, TAG, and TGA) are less likely to occur by chance than in the non-coding regions of A + T-rich genomes. Indeed, the above standard and stringent annotation protocols applied to the A + T-rich (74.8%) *Megavirus chilensis* genome[3] resulted in two very similar sets of predicted versus validated protein-coding genes (1120 versus 1108). This control indicates that our stringent annotation is not simply discarding eventually correct gene predictions by arbitrary raising a confidence threshold, but specifically correcting errors induced by the G + C-rich composition. Purely computational gene annotation methods are thus markedly less reliable for G + C-rich genomes, especially when they encode a large proportion of ORFans (i.e., ORF without database homolog), as for pandoraviruses. However, it is worth noticing that even after our stringent reannotation, the fraction of predicted proteins without significant sequence similarity outside of the *Pandoraviridae* family remained quite high (from 67 to 73%, Supplementary Fig. 4).

An additional challenge for the accurate annotation of the pandoravirus genomes is the presence of introns (virtually undetectable by computational methods when they interrupt ORFans). The mapping of the assembled transcript sequences onto the genomes of *P. salinus*, *P. dulcis*, *P. quercus*, and *P. neocaledonia*, allowed the detection of spliceosomal introns in 7.5–13% of the validated protein-coding genes. These introns were found in the untranslated regions (UTRs) as well as in the coding sequences, including on average 14 genes among those encoding the 200 most abundant proteins detected in the particles (see below). Although spliceosomal introns are found in other viruses with a nuclear phase such as the chloroviruses[9], pandoraviruses are the only ones for which spliceosomal introns have been validated for more than 10% of their genes. These results support our previous suggestion that at least a portion of the pandoravirus transcripts are synthetized and processed by the host nuclear machinery[5]. Yet, the number of intron per viral gene remains much lower (around 1.2 in average) than for the host genes (6.2 in average[10]). Pandoravirus genes also exhibit UTRs twice as long (Supplementary Table 1) as those of *Mimiviridae*[11].

The mapping of the RNA-seq data led to the unexpected discovery of a large number (157–268) of long non-coding transcripts (LncRNAs) (Table 1, Supplementary Table 1 for detailed statistics). These LncRNAs exhibit a polyA tail and about 4% of them contain spliceosomal introns. LncRNAs are most often transcribed from the reverse strand of validated protein-

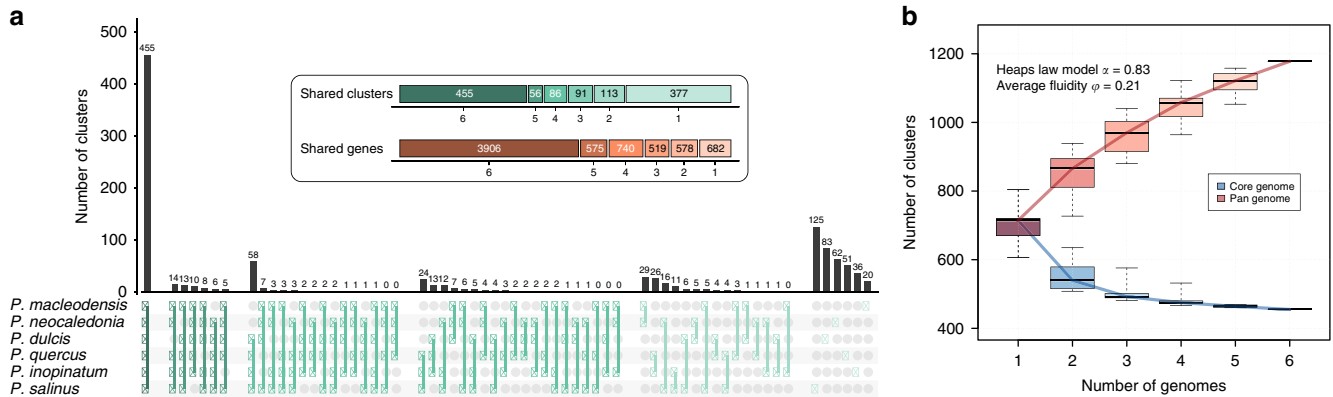

**Fig. 2** Comparison of the pandoravirus gene contents. **a** The distribution of all the combinations of shared protein clusters is shown. The inset summarizes the number of clusters and genes shared by 6, 5, 4, 3, 2, and 1 pandoraviruses. **b** Core genome and pan-genome estimated from the six available pandoraviruses. The estimated heap law $\alpha$ parameter ($\alpha < 1$) is characteristic of an open pan-genome[50] and the fluidity parameter value characteristic of a large fraction of unique genes[51]. Box plots show the median, the 25th, and 75th percentiles. The whiskers correspond to the extreme data points

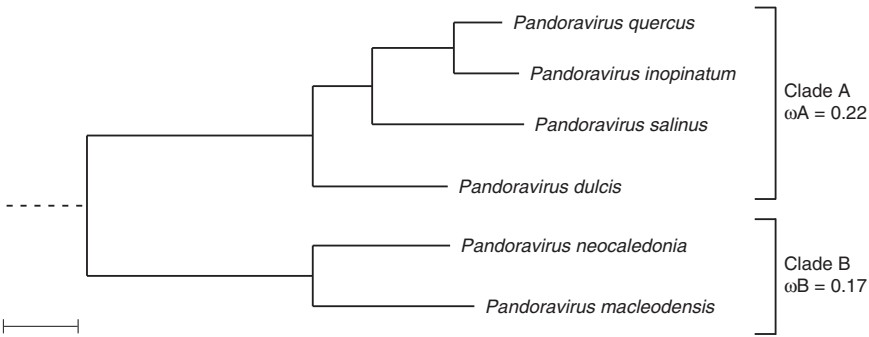

**Fig. 3** Phylogenetic structure of the proposed *Pandoraviridae* family. Bootstrap values estimated from resampling are all equal to 1 and so were not reported. Synonymous to non-synonymous substitution rates ratios ($\omega$) were calculated for the two separate clades and are significantly different (scale bar is 0.07 substitution/site)

coding genes while a smaller fraction are expressed in intergenic (i.e., inter-ORF) regions (Supplementary Fig. 5). These non-coding transcripts may play a role in the regulation of pandoravirus genes expression.

Overall, 82.7–87% of the pandoravirus genomes is transcribed (including ORFs, UTRs, and LncRNAs), but only 62–68.2% is translated into proteins. Such values are much lower than in giant viruses from other families (e.g., 90% of the Mimivirus[11] genome is translated), in part due to the larger UTRs flanking the pandoravirus genes.

**Comparative genomics.** The six protein-coding gene sets obtained from the above stringent annotation were then used as references for whole-genome comparisons aiming to identify specific features of the *Pandoraviridae* family. Following a sequence similarity-based clustering (see Methods), the relative overlaps of the gene contents of the various strains were computed (Fig. 2a), producing what we refer to as "protein clusters".

We then computed the number of shared (i.e., "core") and total genes as we incrementally incorporated the genomes of the various isolates into the above analysis, to estimate the size of the family core gene set and that of the accessory/flexible gene set. If the six available isolates appeared sufficient to delineate a core genome coding for 455 different protein clusters, the "saturation curve" leading to the total gene set is far from reaching a plateau, suggesting that the *Pandoraviridae* pan-genome is open, with each additional isolate predicted to contribute more than 50 additional genes (Fig. 2b). This remains to be confirmed by the analysis of additional *Pandoraviridae* isolates.

We then investigated the global similarity of the six pandoravirus isolates by analyzing their shared gene contents both in term of protein sequence similarity and genomic position. The pairwise similarity between the different pandoravirus isolates ranges from 54 to 88%, as computed from a super alignment of the protein products of the orthologous genes (Supplementary Table 2). A phylogenetic tree computed with the same data clusters the pandoraviruses into two separate clades (Fig. 3).

Interpreted in a geographical context, this clustering pattern conveys two important properties of the emerging family. On one hand, the most divergent strains are not those isolated from the most distant locations (e.g., the Chilean *P. salinus* versus the French *P. quercus*; the Neo-Caledonian *P. neocaledonia* versus the Australian *P. macleodensis*). On the other hand, two isolates (e.g., *P. dulcis* versus *P. macleodensis*) from identical environments (two ponds located 700 m apart and connected by a small water flow) are quite different. Pending a larger-scale inventory of the *Pandoraviridae*, these results already suggest that members of this family are distributed worldwide with similar local and global diversities.

Our analysis of the positions of the homologous genes in the various genomes revealed that despite their sequence divergence (Supplementary Table 2), 80% of the orthologous genes remain collinear. As shown in Fig. 4, the long-range architecture of the pandoravirus genomes (i.e., based on the positions of orthologous genes) is globally conserved, despite their differences in sizes (1.83–2.47 Mb). However, one-half of the pandoravirus chromosomes (the leftmost region in Fig. 4) curiously appears

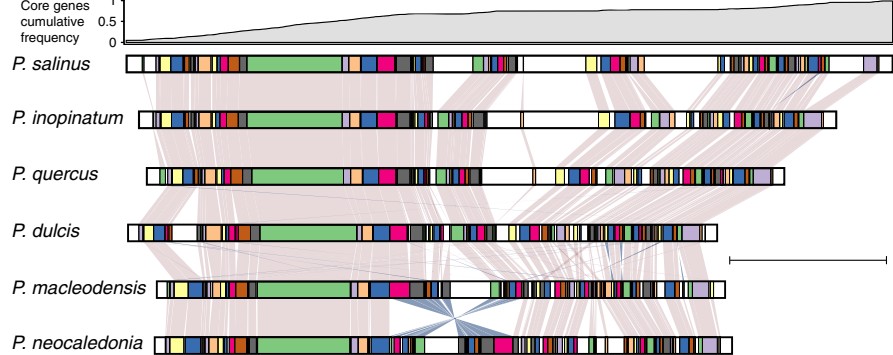

**Fig. 4** Collinearity of the available pandoravirus genomes. Cumulative frequency of core genes is shown at the top. Conserved collinear blocks are colored in the same color in all viruses. White blocks correspond to non-conserved DNA segments (scale bar is 500 kb)

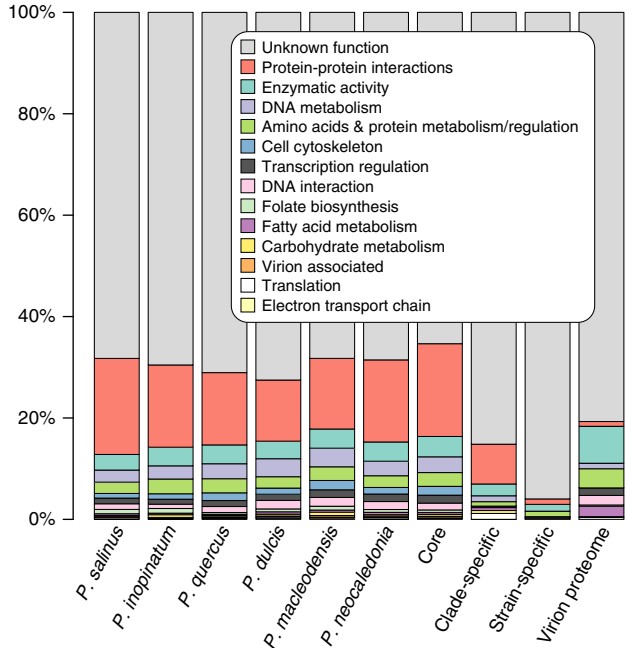

**Fig. 5** Functional annotations

evolutionary more stable than the other half where most of the non-homologous segments occur. These segments contain strain-specific genes and are enriched in tandem duplications of non-orthologous ankyrin, MORN, and F-box motif-containing proteins. Conversely, the stable half of the genome concentrates most of the genes constituting the *Pandoraviridae* core genome (top of Fig. 4). Interestingly, the local inversion that distinguishes the chromosome of *P. neocaledonia* from the other strains is located near the boundary between the stable and unstable regions, and may be linked to this transition (although it may be coincidental). Finally, all genomes are also enriched in strain-specific genes (and/or duplications) at both extremities.

We then analyzed the distribution of the predicted proteins among the standard broad functional categories (Fig. 5). As it is now recurrent for large and giant eukaryotic DNA viruses, the dominant category is by far that of proteins lacking recognizable functional signatures. Across the six strains, an average of 70% of the predicted proteins correspond to "unknown functions". Such a high proportion is all the more remarkable as it applies to carefully validated gene sets, from which dubious ORFs have been eliminated. It is thus a biological reality that a large majority of these viral proteins cannot be linked to previously characterized pathways. Remarkably, the proportion of such anonymous

proteins remains quite high (65%) among the products of the pandoravirus core genome, that is among the presumably essential genes shared by the six available strains (and probably all future family members, according to Fig. 2b). Interestingly, this proportion remains also very high (≈80%) among the proteins detected as constituting the viral particles. Furthermore, the proportion of anonymous proteins totally dominates the classification of genes unique to each strain, at more than 95%. The most generic functional category, "protein–protein interaction" is the next largest (from 11.7 to 18.9%), corresponding to the detection of highly frequent and uninformative motifs (e.g., ankyrin repeats). Overall, the proportion of pandoravirus proteins to which a truly informative function could be attributed is <20%, including a complete machinery for DNA replication and transcription.

We then investigated two evolutionary processes possibly at the origin of the extra-large size of the pandoravirus genomes: horizontal gene transfers (HGTs) and gene duplications. The acquisition of genes by HGT was frequently invoked to explain the genome size of ameba-infecting viruses compared to "regular" viruses[12, 13]. We computed that up to a third of the pandoravirus proteins exhibit sequence similarities (outside of the *Pandoraviridae* family) with proteins from the three cellular domains (Eukarya, Archaea, and Eubacteria) or other viruses (Supplementary Fig. 4). However, such similarities do not imply that these genes were horizontally acquired. They also could denote a common ancestral origin or a transfer from a pandoravirus to other microorganisms. We individually analyzed the phylogenetic position of each of these cases to infer their likely origin: ancestral —when found outside of clusters of cellular or viral homologs; horizontally acquired—when found deeply embedded in the above clusters; or horizontally transferred to cellular organisms or unrelated viruses in the converse situation (i.e., a cellular protein lying within a pandoravirus protein cluster). Supplementary Fig. 6 summarizes the results of this analysis.

We could make an unambiguous HGT diagnosis for 39% of the cases, the rest remaining undecidable or compatible with an ancestral origin. Among the likely HGT, 49% suggested a horizontal gain by pandoraviruses, and 51% the transfer of a gene from a pandoravirus. Interestingly, the acquisition of host genes, a process usually invoked as important in the evolution of viruses, only represent a small proportion (13%) of the diagnosed HGTs, thus less than from the viruses to the host (18%). Combining the above statistics with the proportion of genes (one-third) we started from, in the whole genome, suggests that at most 15% (and at least 6%) of the pandoravirus gene content could have been gained from cellular organisms (including 5–2% from their contemporary *Acanthamoeba* host) or other viruses. Such range of values is comparable to what was previously estimated

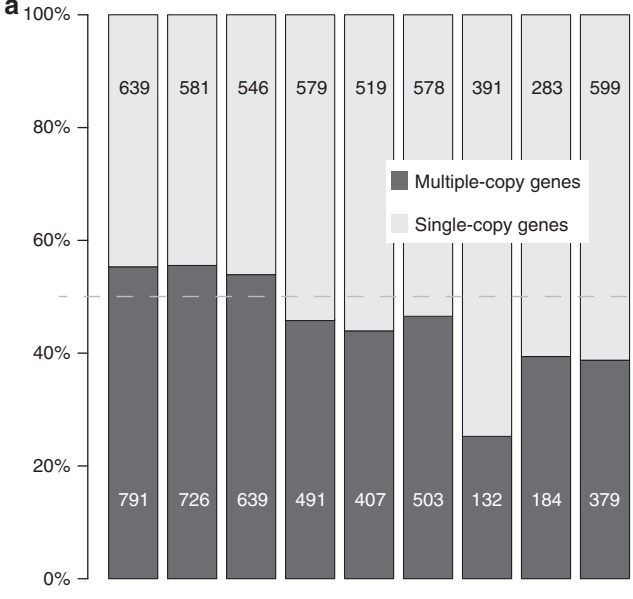

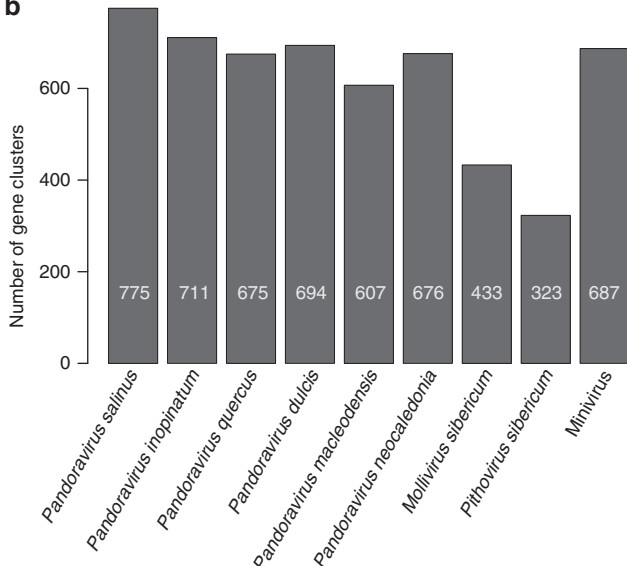

**Fig. 6** Analysis of gene duplication in various giant virus families. **a** Distribution of single-copy versus multiple-copy genes in giant viruses. **b** Number of distinct gene clusters

genes in pandoraviruses is similar to, and sometimes smaller (e.g., *P. neocaledonia*, 2 Mb) than that in Mimivirus, with a genome (1.18 Mb) half the size. Overall, the number of distinct gene clusters (Fig. 6b) overlaps between the *Pandoraviridae* (from 607 to 775) and Mimivirus (687), suggesting that despite their difference in genome and particle sizes, these viruses share comparable genetic complexities.

Gene duplication being such a prominent feature of the pandoravirus genomes, we investigated it further looking for more insight about its mechanism. First, we computed the genomic distances between pairs of closest paralogs, most likely resulting from the most recent duplication events. The distributions of these distances, similar for each pandoravirus, indicate that the closest paralogs are most often located next to each other (distance = 1) or separated by a single gene (distance = 2) (Supplementary Fig. 8).

We then attempted to correlate the physical distance separating duplicated genes with their sequence divergence as a (rough) estimate of their evolutionary distance. We obtained a significant correlation between the estimated "age" of the duplication event and the genomic distance of the two closest paralogs (Supplementary Fig. 9). These results suggest an evolutionary scenario whereby most duplications are first occurring in tandem, with subsequent genome alterations (insertions, inversions, and gene losses) progressively blurring this signal.

**Comparative proteomic of pandoravirions**. Our previous mass spectrometry proteomic analysis of *P. salinus* particles identified 210 viral gene products, most of which ORFans or without predictable function. In addition, we detected 56 host (Acantamoeba) proteins. Importantly, none of the components of the virus-encoded transcription apparatus was detected in the particles[5]. In this work we performed the same analyses on *P. salinus*, *P. dulcis*, and two of the new isolates (*P. quercus* and *P. neocaledonia*) to determine to what extent the above features were conserved for members of the *Pandoraviridae* family with various levels of divergence, and identify the core versus the accessory components of a generic pandoravirion.

Due to the constant sensitivity improvement in mass spectrometry, our new analyses of purified virions led to the reliable identification of 424 proteins for *P. salinus*, 357 for *P. quercus*, 387 for *P. dulcis*, and 337 for *P. neocaledonia* (see Methods). However, this increased number of identifications corresponds to abundance values (intensity-based absolute quantification, iBAQ) spanning more than five orders of magnitude. Many of the proteins identified in the low abundance tail might thus not correspond to bona fide particle components, but to randomly loaded bystanders, "sticky" proteins, or residual contaminants from infected cells. This cautious interpretation is suggested by several observations:

– the low abundance tail is progressively enriched in viral proteins identified in the particles of a single pandoravirus strain (even though other strains possess the homologous genes),
– the proportion of host-encoded proteins putatively associated to the particles increases at the lowest abundances,
– many of these host proteins were previously detected in particles of virus unrelated to the pandoraviruses but infecting the same host,
– these proteins are abundant in the Acanthamoeba proteome (e.g., actin, peroxidase, etc) making them more likely to be retained as purification contaminants.

Unfortunately, the iBAQ value distributions associated to the pandoravirion proteomes did not exhibit a discontinuity that

for Mimivirus[14]. HGT is thus not the distinctive process at the origin of the giant pandoravirus genomes.

We then investigated the prevalence of duplications among pandoravirus genes. Figure 6a compares the proportions of single versus duplicated (or more) protein-coding genes of the six available pandoraviruses with that computed for representatives of the three other known families of giant DNA viruses infecting Acanthamoeba. It clearly shows that the proportion of multiple-copy genes (ranging from 55 to 44%) is higher in pandoraviruses, than for the other virus families, although it does not perfectly correlate with their respective genome sizes. The distributions of cluster sizes among the different pandoravirus strains are similar. Most multiple-copy genes are found in cluster of size 2 (duplication) or 3 (triplication). The number of bigger clusters then decreases with their size (Supplementary Fig. 7).

Fewer large clusters (size > 20) correspond to proteins sharing protein–protein interaction motifs, such as Ankyrin, MORN, and F-box repeats. Surprisingly, the absolute number of single-copy

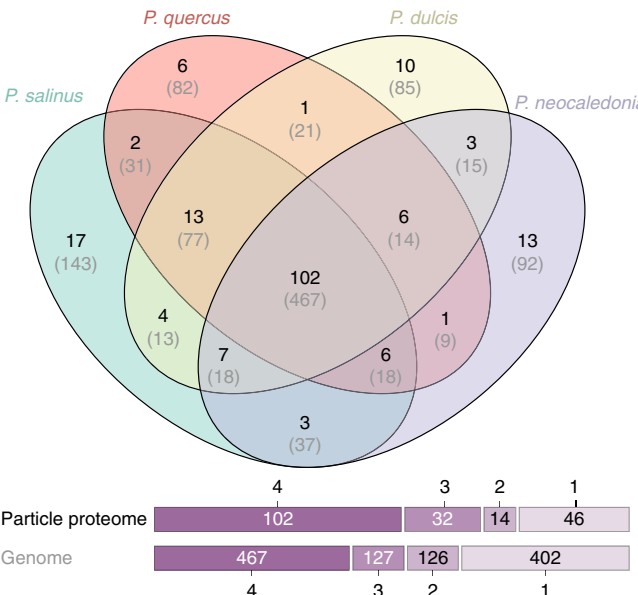

**Fig. 7** Venn diagram of the particle proteomes of four different pandoravirus strains

could serve as an objective abundance threshold to distinguish bona fide particle components from dubious ones. However, the number of identified Acanthamoeba proteins sharply increases after rank ≈200 in the whole proteome (Supplementary Fig. 10). Following the same conservative attitude as for the genome reannotation, we decided to disregard the proteins identified below this rank as likely bystanders and only included the 200 most abundant proteins in our further analyses of the particle proteomes (Supplementary Data 1, Supplementary Table 3). Using this stringent proteome definition for each of the four different pandoravirions, we first investigated the diversity of their constituting proteins and their level of conservation compared to the global gene contents of the corresponding pandoravirus genomes.

Figure 7 shows that the particle proteomes include proteins belonging to 194 distinct clusters, 102 of which are shared by the four strains. The core proteome is thus structurally and functionally diverse. It corresponds to 52.6% of the total protein clusters globally identified in all pandoravirions. By comparison, the 467 protein clusters encoded by the core genome only represents 41.6% (i.e., 467/1122) of the overall number of pandoravirus-encoded protein clusters. The pandoravirus "box" used to propagate the genomes of the different strains is thus significantly more conserved than their gene contents ($p \ll 10^{-3}$, chi-square test). The genes encoding the core proteome also exhibit the strongest purifying selection among all pandoravirus genes (Supplementary Fig. 11a).

To evaluate the reliability of our proteome analyses we compared the abundance (iBAQ) values determined for each of the 200 most abundant proteins for two technical replicates and for two biological replicates performed on the same pandoravirus strain (Supplementary Fig. 12a & b). A very good correlation (Pearson's $R > 0.97$) was obtained in both cases for abundance values ranging over three orders of magnitude. We then compared the iBAQ values obtained for orthologous proteins shared by the virion proteomes of different isolates. Here again, a good correlation was observed ($R > 0.81$), as expected smaller than for the above replicates (Supplementary Fig. 12c & d). These results suggest that although the particles of the different strains appear morphologically identical (Supplementary Fig. 1), they

admit a tangible flexibility both in terms of the protein sets they are made of (with 89% of pairwise orthologues in average), and in their precise stoichiometry.

We then examined the predicted functions of the proteins composing the particles, from the most to the least abundant, hoping to gain some insights about the early infectious process. Unfortunately, only 19 protein clusters could be associated to a functional/structural motif out of the 102 different clusters defining the core particle proteome (Supplementary Data 1, Supplementary Table 3). This proportion is less than for the whole genome (Fig. 5), confirming the alien nature of the pandoravirus particle as already suggested by its unique morphology and assembly process[5]. The pandoravirions are mostly made of proteins without homologs outside of the *Pandoraviridae* family. No protein even remotely similar to the usually abundant major capsid protein (MCP), a predicted DNA-binding core protein, or a DNA-packaging ATPase, hallmarks of most eukaryotic large DNA viruses, is detected. In particular, a *P. salinus* hypothetical protein (previously ps_862 now reannotated psal_cds_450) recently suggested by Sinclair et al.[15] to be a strong MCP candidate was not detected in the *P. salinus* virions, nor its homologs in the other strain proteomes. This negative result emphasizes the need for the experimental validation of computer predictions made from the "twilight zone" of sequence similarity. No trace of the pandoravirus-encoded RNA polymerase is detected either, confirming that the initial stage of infection requires the host transcription machinery located in the nucleus. Spliceosomal introns were validated for 56 pandoravirus genes the products of which were detected in the pandoravirions (Supplementary Data 1). This indicates the preservation of a functional spliceosome until the end of the infectious cycle, as expected from the observation of unbroken nuclei (Supplementary Fig. 1).

Among the 19 non-anonymous protein clusters, 4 exhibit generic motifs without specific functional clue: 2 collagen-like domains and 1 Pan/APPLE-like domain that are involved in protein–protein interactions, and 1 cupin-like domain corresponding to a generic barrel fold. Among the 10 most abundant core proteins, 9 have no predicted function, except for 1 exhibiting a C-terminal thioredoxin-like domain (psal_cds_383). It is worth noticing that the predicted membrane-spanning segment of 22 amino acids (85–107) is conserved in all pandoravirus strains. The 5′UTR of the corresponding genes exhibit 2 introns (in *P. salinus*, *P. dulcis*, and *P. quercus*) and 1 in *P. neocaledonia*. Thioredoxin catalyzes dithiol-disulfide exchange reactions through the reversible oxidation of its active center. This protein, with another one of the same family (psal_cds_411, predicted as soluble), might be involved in repairing/preventing phagosome-induced oxidative damages to viral proteins prior to the initial stage of infection. The particles also share another abundant redox enzyme, an ERV-like thiol oxidoreductase that may be involved in the maturation of Fe/S proteins. Another core protein (psal_cds_1260) with a remote similarity to a thioredoxin reductase may participate to the regeneration of the oxidized active sites of the above enzymes. Among the most abundant core proteins, psal_cds_232 is predicted as DNA-binding, and may be involved in genome packaging. One putative NAD-dependent amine oxidase (psal_cds_628), and one FAD-coupled dehydrogenase (psal_cds_1132) complete the panel of conserved putative redox enzymes. Other predicted core proteins include a Ser/thr kinase and phosphatase that are typical regulatory functions. One serine protease, one lipase, one patatin-like phospholipase, and one remote homolog of a nucleoporin might be part of the toolbox used to ferry the pandoravirus genomes to the cytoplasm and then to the nucleus (Supplementary Table 3). Finally, two core proteins (psal_cds_118 and psal_cds_874) share an

endoribonuclease motif and could function as transcriptional regulators targeting cellular mRNA.

At the opposite of defining the set of core proteins shared by all pandoravirions, we also investigated strain-specific components. Unfortunately, most of the virion proteins unique to a given strain (about 10 in average) are anonymous and in low abundance. No prediction could be made about the functional consequence of their presence in the particles.

## Discussion

We isolated three new pandoraviruses (*P. neocaledonia*, *P. quercus*, and *P. macleodensis*) from distant locations (resp. New Caledonia, South of France, and Australia). As for the previously characterized members of this emerging family (*P. salinus* from Chile, *P. dulcis* from Australia, and *P. inopinatum* from Germany), their genomes consist in large linear G + C-rich dsDNA molecules around 2 Mb in size (Table 1). Using the four most divergent pandoravirus strains at our disposal, we combined RNA-seq, virion proteome, and sequence similarity analyses to design a stringent annotation procedure, eliminating most false positive gene predictions that could both inflate the proportion of ORFans and distort the results of comparative analyses. Our—probably over-cautious—gene-calling procedure (reducing the number of predicted proteins by up to 44%) (Table 1) nevertheless confirmed that an average of 70% of the experimentally validated genes encode proteins without detectable homolog outside of the *Pandoraviridae* family, and up to 80% for those detected in the particles.

Using the six strains known as of today, we determined that each new family member contributed genes not previously seen in the other genomes, at a rate suggesting that the *Pandoraviridae* pan-genome is open (Fig. 2). Moreover, this flexible (i.e., strain- and clade-specific) gene subset exhibits a much higher proportion of ORFans (respectively 96% and 90%) than the core genome (63%) (Supplementary Fig. 11d). According to the usual interpretation, the core genome corresponds to genes inherited from the last common ancestor of a group of viruses while the flexible genome corresponds to genes that appeared since their divergence, through various mechanisms. We then performed further comparative statistical analyses to investigate which mechanisms might be responsible for the large pandoravirus gene content and, possibly, of its continuous expansion.

We determined that gene duplication was a contributing factor in the genome size of pandoraviruses, with 50% of their genes present in multiple copies (Fig. 6, Supplementary Fig. 7). However, this value is not vastly different from the proportion (40%) computed for Mimivirus with a genome half the size (Fig. 6). Thus, duplication alone does not explain the much larger gene content of the pandoraviruses. The proportion of single-copy ORFans (from 50.7 to 62.7%) compared to those in multiple copies is significantly larger than for non-ORFans (from 30 to 44.5%) (Fischer exact test, *p*-value < $2 \times 10^{-4}$). ORFan genes thus tend to be less frequently duplicated.

HGT is also frequently invoked as a mechanism for viral genome inflation[12,13,16–18]. We estimated that HGT might be responsible for 6 to 15% of the *P. salinus* gene content (Supplementary Fig. 6). Such a proportion is not exceptional compared to other large eukaryotic dsDNA viruses[14] with much smaller genomes, and thus does not explain the huge pandoravirus gene content. Furthermore, the large proportion of ORFans among the flexible genome (Supplementary Fig. 11) is arguing against recent acquisitions from HGTs, short of postulating that they originated from mysterious organisms none of which has yet been characterized. Alternatively, the phylogenetic signal from these newly acquired genes could have been erased due to accelerated

evolution. However, this is not supported by our data showing that pandoravirus-specific ORFan genes are under strong purifying selection, just to a lesser extent than non-ORFans (Supplementary Fig. 11).

The proportion of ORFans (i.e., proteins without homologs in the databases) obviously depends on our limited knowledge of the virosphere. However, what characterizes the *Pandoraviridae* is the unprecedented number of family-specific ORFans they share, the increase of their proportion among the subsets of core genes (with orthologs in all strains), clade-specific and strain-specific genes (Supplementary Fig. 11d), as well as their distinctive statistical properties (Supplementary Fig. 13). Altogether, this suggests that the pandoravirus-specific ORFans are not just ancestral genes missing from the database, but genes with histories confined within the *Pandoraviridae*.

To further investigate the origin of the pandoravirus genes, we performed various statistical analyses in search of what would distinguish core genes from clade-specific genes, and from those unique to each strain. To ensure the assignation of each of the genes to their respective categories, we added a constraint on their genomic positions. For instance, we only considered strain-specific genes found interspersed within otherwise collinear sequences of clade-specific or core genes (Fig S13a). The genes from the three above categories appeared significantly different with respect to three independent properties (G + C content, ORF length, and CAI). Moreover, the clade-specific and strain-specific genes exhibited average values intermediate between that of the core genes and intergenic sequences (Supplementary Fig. 13b-d). Such a gradient unmistakably advocates what is referred to as the de novo protein creation (reviewed in refs. [19–22]). Our data support an evolutionary scenario whereby novel (hence strain-specific) protein-coding genes could randomly emerge from non-coding intergenic regions, then become alike protein-coding genes of older ancestry (i.e., clade-specific and core genes) in response to an adaptive selection pressure (Supplementary Fig. 11b). For a long time considered unrealistic on statistical ground[23], the notion that new protein-coding genes could emerge de novo from non-coding sequences[24] started to gain an increasing support following the discovery of many expressed ORFan genes in *Saccharomyces cerevisiae*[25], *Drosophila*[26], *Arabidopsis*[27], mammals[28], and primates[29]. This hypothesis was recently extended to giant viruses[30].

A different process, called overprinting, involves the use of alternative translation frame from preexisting coding regions. It appears mostly at work in small (mostly RNA) viruses and bacteria, the dense genomes of which lacks sufficient non-coding regions[31, 32]. However, overprinting would not generate the observed difference in G + C composition between strain-specific and core genes (Supplementary Fig. 13b).

The eukaryotic-like de novo gene creation hypothesis might apply to the pandoraviruses for several reasons. This process requires ORFs that are abundant (to compensate for its contingent nature) and large enough (e.g., >150 bp) to encode peptides capable of folding into minimal domains (40–50 residues). We previously pointed out that the high G + C content of the pandoraviruses, compared to the A + T richness of the other Acanthamoeba-infecting viruses[8], statistically increases the size of the random ORFs in non-coding regions. Moreover, these non-coding regions are also larger in average, representing up to 38% of the total genome (Supplementary Table 1). The pandoravirus genomes thus offer an ideal playground for de novo gene creation. However, a high G + C composition does not imply viral genome inflation and/or an open-ended flexible gene content, as shown by Herpesviruses, another family of dsDNA virus replicating in the nucleus[33]. Even though HSV-1 and HSV-2 exhibit a G + C content of 68% and 70% respectively, their genome

remained small (≈150 kb), and their genes coding for core proteins, non-core proteins, as well as their relatively large intergenic regions (≈250 ± 150 bp) do not display any significant difference in composition[34]. Accordingly, a single gene (US12) has been suggested to have emerged de novo[35]. Thus, pandoraviruses (and/ or their amoebal host) must exhibit some specific features leading them to favor de novo gene creation. This might be the extensible genome space offered in their particle, the uncondensed state of their DNA genome, the absence of DNA repair enzymes packaged in the virion, or an unknown template-free machinery generating new DNA. The later mechanism, although highly speculative, would be easier to reconcile with the conserved collinearity of the pandoravirus genome than intense mutagenesis, duplication, or the shuffling of preexisting genes. This template-free generation process might be linked to the apparent instability of the right half of the pandoravirus chromosome, depleted in "core genes" (Fig. 4). We need more genomes to validate the bipartite heterogeneity of the pandoravirus chromosomes as a distinctive property of the family.

Conceptually, de novo gene creation can occur in two different ways: an intergenic sequence gains transcription before evolving an ORF, or the converse[21]. The numerous LncRNAs that we detected during the infection cycle of the various pandoraviruses would appear to favor the transcription first mechanisms. However, most of these non-coding transcripts are antisense of bona fide coding regions and would not generate the shift in G + C composition observed for strain-specific genes. Novel proteins might thus mostly emerge from the numerous intergenic (random) ORFs gaining transcription.

The best evidence of de novo gene creation, although rarely obtained, is the detection of a significant similarity between the sequence encoding a strain-specific ORFan protein and an intergenic sequence in a closely related strain[19]. Out of the 318 pandoravirus strain-specific genes that we tested, we found two of such occurrences. The P. salinus psal_cds_1065 (58 aa, 55% GC, CAI = 0.287) is similar to a non-coding RNA (pneo_ncRNA_241) in P. neocaledonia, and the P. salinus psal_cds_415 (96 aa, 54% GC, CAI = 0.173) is matching within an intergenic region in P. quercus. In both cases, the matches occur at homologous genomic location. Such low rate of success (yet a positive proof of principle) was expected given the sequence divergence of the available pandoravirus strains, especially in their intergenic regions.

If we now admit the hypothesis that de novo gene creation plays a significant role in the large proportion of strain-specific ORFans and in the open-ended nature of the Pandoraviridae gene content, it could also have contributed to the pool of family-specific ORFans genes (now shared by two to six strains) to an unknown extant. The nature of the ancestor of the Pandoraviridae thus remains an unresolved question. Invoking the de novo creation hypothesis greatly alleviates the problem encountered when attempting to explain the diversity of the Pandoraviridae gene contents by lineage-specific gene losses and reductive evolution[8]. Instead of postulating an increasingly complex ancestor as new isolates are exhibiting additional unique genes, we can now attribute them to de novo creation. Yet, lineage-specific losses can still account for the gene content partially shared among strains.

As seductive as it is, the de novo creation hypothesis is nevertheless plagued by its own difficulties. First, newly expressed (random) proteins have to fold in a compact manner, or at least in a way not interfering with established protein interactions. Although early theoretical studies suggested that stable folding of random amino-acid sequences might be improbable[36], several experimental studies have indicated success rate of up to 20% [37], [38]. It has also been suggested that proteins encoded by de novo-created genes might be enriched in disordered regions[39].

Accordingly, we observed a slightly albeit significant ($p < 10^{-15}$, Wilcoxon signed-rank test) higher fraction of predicted disordered residues[40] in ORFans (14%) versus non-ORFans (11%). Also challenging is the process by which a random protein would spontaneously acquire a function. For example, only four functional (ATP-binding) proteins resulted from the screening of $6 \times 10^{12}$ random sequences followed by many iterations of in vitro selections and directed evolution[41]. At the same time, the spontaneous mutation rate of large dsDNA viruses is very low (estimated at $<10^{-7}$ substitution per position per infection cycle)[42]. In absence of a useful function on which to exert a purifying selection, it seems very unlikely that a newly created protein could remain in a genome long enough to acquire a selectable influence on the virus fitness. How the so-called protogene[25] manage to be retained through the intermediate steps eventually leading to a selectable function remains the dark part of any de novo gene creation scenario. Thus, if our comparative genomic studies suggest new hypotheses about the evolution of pandoraviruses and other giant amoebal viruses, it is far from closing the debate about the genetic complexity of their ancestor[8, 17, 18, 43, 44].

In the context of this debate, it was previously proposed[17] that the pandoraviruses were highly derived phycodnaviruses based on the phylogenetic analysis of a handful of genes while disregarding the amazingly unique structural and physiological features displayed by the first two pandoravirus isolates[5] as well as the huge number of genes unique to them. Now using the six available pandoravirus genomes, a cladistic clustering based on the presence/absence of homologous genes in the different virus groups robustly separates the proposed Pandoraviridae family from the previously established families of large eukaryote-infecting dsDNA viruses (Fig. 8). The only remaining uncertainty concerns the actual position of the yet unclassified Mollivirus sibericum virus that will eventually be the seed of a distinct viral family, or the prototype of an early diverging branch[45] of smaller pandoraviruses. More Pandoravidae members are needed to delineate the exact boundaries of this new family and resolve the many issues we raised about the origin and mode of evolution of its members.

## Methods

**Environmental sampling and virus isolation**. P. neocaledonia: A sample from the muddy brackish water of a mangrove near Noumea airport (New Caledonia, Lat: 22°16′29.50″S, Long: 166°28′11.61″E) was collected. After mixing the mud and the water, 50 mL of the solution was supplemented with 4% of rice media (supernatant obtained after autoclaving 1 L of seawater with 40 grains of rice) and let to incubate in the dark. After 1 month, 1.5 mL were recovered and 150 μL of pure Fungizone (25 μg/mL final) was added to the sample, which was vortexed and incubated overnight at 4 °C on a stirring wheel. After sedimentation, 1 mL supernatant was recovered and centrifuged at 800 × g for 5 min. Acanthamoeba A. castellanii (Douglas) Neff (ATCC 30010TM) cells adapted to Fungizone (2.5 μg/mL) were inoculated with 100 μL of the supernatant as previously described[46] and monitored for cell death.

P. macleodensis: A muddy sample was recovered from a pond 700 m away but connected to the La Trobe University pond in which P. dulcis was isolated[5]. After mixing the mud and the water, 20 mL of sample were passed through a 20 μm sieve and the filtrate was centrifuged 15 min at 30 000 × g. The pellet was resuspended in 200 μL of phosphate-buffered saline (PBS) supplemented with antibiotics and 30 μL was added to six wells of culture of A. castellanii cells adapted to Fungizone (see SI Materials and Methods).

P. quercus: Soil under decomposing leaves was recovered under an oak tree in Marseille. Few grams were resuspended with 12 mL PBS supplemented with antibiotics. After vortexing 10 min, the tube was incubated during 3 days at 4 °C on a stirring wheel. The tube was than centrifuged 5 min at 200 × g and the supernatant was recovered, centrifuged 45 min at 6800 × g. The pellet was resuspended in 500 μL PBS supplemented with the antibiotics. A volume of 50 μL of supernatant and 20 μL of the resuspended pellet were used to infect A. castellanii cells adapted to Fungizone. As for P. neocaledonia and P. macleodensis, visible particles resembling pandoraviruses were visible in the culture media after cell lysis. All viruses were then cloned using a previously described procedure[45] prior to DNA extraction for sequencing and protein extraction for proteomic studies.

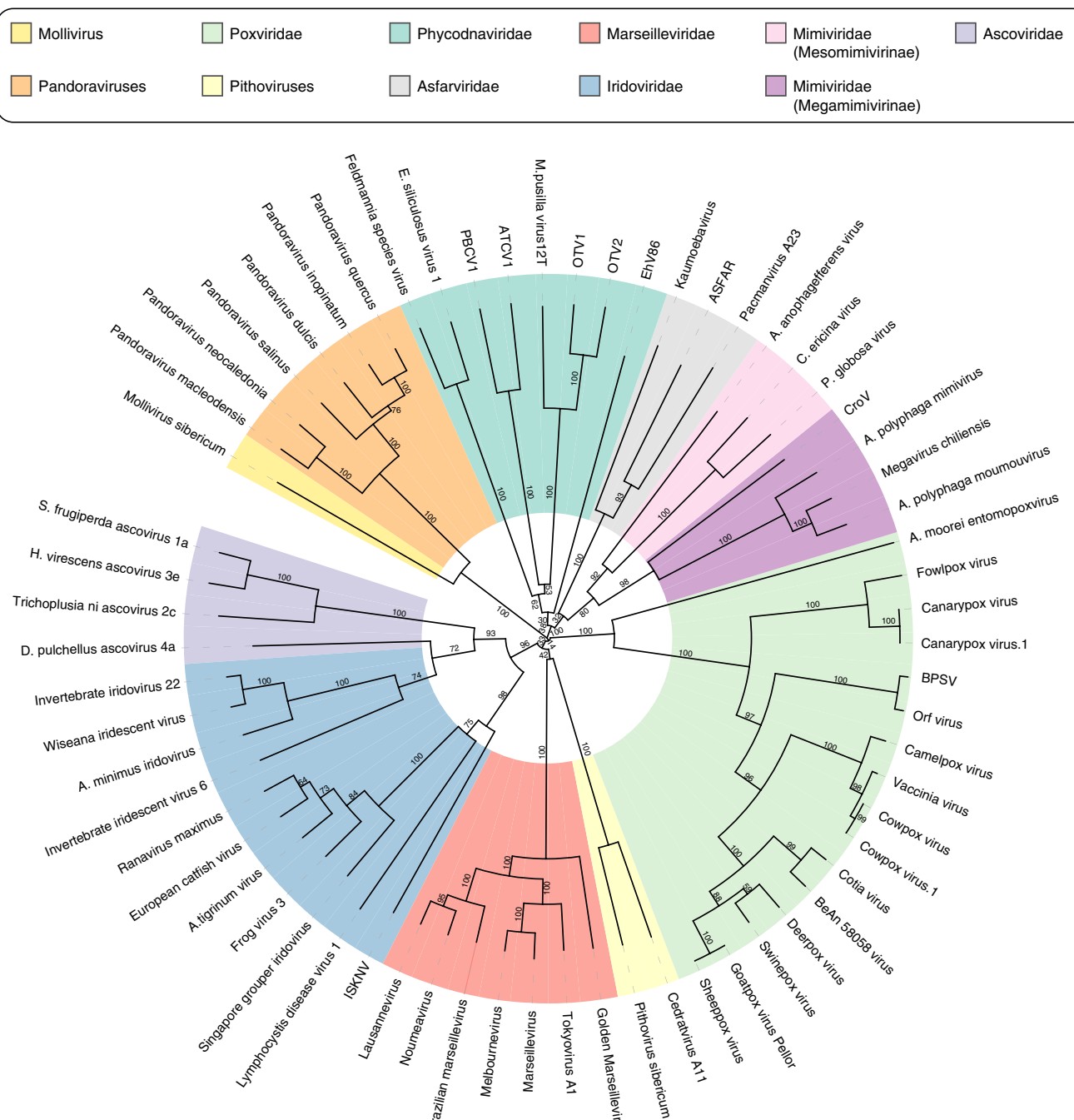

**Fig. 8** Gene content-based cladistic tree of large DNA viruses. Long virus names have been replaced by acronyms (from top, clockwise). OTV1 Ostreococcus tauri virus 1, OTV2 Ostreococcus tauri virus 2, EhV86 Emiliania huxleyi virus 86, ASFAR African swine fever virus, CroV Cafeteria roenbergensis virus BV.PW1, BPSV Bovine papular stomatitis virus, ISKNV Infectious spleen and kidney necrosis virus. A maximum likelihood phylogenetic tree based on the DNA polymerase B protein sequence showing a globally similar topology was also computed (Supplementary Fig. 14)

Synchronous infections were performed for transmission electron microscopy observations of the infectious cycles. mRNA were extracted from the pooled infected cells prior to polyA+ enrichment and sent for library preparation and sequencing.

**Genome sequencing and assembly.** *P. neocaledonia* and *P. quercus* genomes were sequenced using the Pacbio sequencing technology. *P. macleodensis* genome was sequenced using the Illumina MiSeq technology with large insert (5–8 kb) mate pair sequences. Details on the strategy used for the genome assemblies are provided in the SI Materials and Methods.

**Genome sequence stringent annotation.** A stringent genome annotation was performed using a combination of ab initio gene prediction, strand-specific RNA-

seq transcriptomic data, mass spectroscopy proteomic data as well as protein conservation data. The pipeline used is summarized in Supplementary Fig. 2 and described in the SI Materials and Methods.

**Proteomic analyses.** Virion proteomes were prepared as previously described in ref. [45] for mass spectrometry-based label-free quantitative proteomics. Briefly, extracted proteins from each preparation were stacked in the top of a 4–12% NuPAGE gel (Invitrogen) before R-250 Coomassie blue staining and in-gel digestion with trypsin (sequencing grade, Promega). Resulting peptides were analyzed by online nanoLC-MS/MS (Ultimate 3000 RSLCnano and Q-Exactive Plus, Thermo Scientific) using a 120-min gradient. Three independent preparations from the same clone were analyzed for each pandoravirus to characterize particle composition. Characterization of different clones and technical replicates were

performed for *P. dulcis*. Peptides and proteins were identified and quantified as previously described[45] (SI Materials and Methods).

**Miscellaneous bioinformatic analyses**. A detailed description of the bioinformatics analyses used for protein clustering and genome rearrangements is detailed in the SI Materials and Methods. CAI was measured using the cai tool from the EMBOSS package[47]. The reference codon usage was computed from the *A. castellanii* most expressed genes (Supplementary Data 2). DNA-binding prediction of pandoravirus proteins was computed using the DNABIND server[48].

**Data availability**. The annotated genomic sequence determined for this work as well as the reannotated genomic sequences have been deposited in the Genbank/EMBL/DDBJ database under the following accession numbers: *P. salinus*, KC977571; *P. dulcis*, KC977570; *P. quercus*, MG011689; *P. neocaledonia*, MG011690; and *P. macleodensis*, MG011691. The reannotated *P. inopinatum* genome used in our comparative analyses is provided as Supplementary Data 3.

The mass spectrometry proteomics data have been deposited to the ProteomeXchange Consortium via the PRIDE[49] partner repository with the dataset identifier PXD008167.

All the genomic data, gene annotations, and transcriptomic data can be visualized on an interactive genome browser at the following address: [http://www.igs.cnrs-mrs.fr/pandoraviruses/]. All data are available from the authors.

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

## Acknowledgements

This work was partially supported by the French National Research Agency (ANR-14-CE14-0023-01), France Genomique (ANR-10-INSB-01-01), Institut Français de Bioinformatique (ANR–11–INSB–0013), the Fondation Bettencourt-Schueller (OTP51251), by a DGA-MRIS scholarship, and by the Provence-Alpes-Côte-d'Azur région (2010 12125). Proteomic experiments were partly supported by the Proteomics French Infrastructure (ANR-10-INBS-08-01) and Labex GRAL (ANR-10-LABX-49-01). We thank the support of the discovery platform and informatics group at EDyP. We thank Deborah for her thorough reading of the manuscript.

## Author contributions

M.L., Y.C., C.A., and J.-M.C. conceived and designed the research; E.F., S.J., A.L., J.-M.A., L. Beucher, N.P., L. Bertaux, E.C.-F., and Y.C. performed experimental research; M.L., O. P., C.A., and J.-M. C. performed bioinformatic analyses; L. Beucher, K.L., and Y.C. contributed new reagents/analytic tools; M.L., C.A., and J.-M.C. analyzed data; M.L., C. A., and J.-M.C. wrote the paper.

## Additional information

**Competing interests:** The authors declare no competing interests.

