## [Peer Review File · Nature Communications]

Reviewers' comments:

Reviewer #1 (Remarks to the Author):

In this highly interesting and data-loaded contribution, Legendre and coworkers describe three new Pandoravirus isolates and compare them to three previously published isolates. By comparing genomic features, transcriptomes, and particle proteomes, the authors generate a comprehensive feature list for the emerging family Pandoraviridae.

The authors conduct a stringent gene annotation procedure based on sequence homology and experimental validation, which leads to a greatly reduced number of putative protein-coding genes, but also to the discovery of new non-coding RNAs and spliceosomal introns. The original inflated gene numbers for Pandoraviruses are thus due to erroneous annotation of small ORFs caused by the paucity of stop codons in GC-rich genomes.

Pandoraviruses have generated great interest due to their extremely large genomes and it is therefore highly relevant that this paper reveals that after applying stringent genome annotation methods, the genetic complexity of pandoraviruses is quite comparable to the significantly smaller mimiviruses. According to the authors' analyses, gene duplication and HGT events contributed to the size of the pandoravirus genome, although not exclusively, and in the case of HGT, not following the stereotypical one-way host-to-virus route. The authors suggest that de novo gene creation might also be an important factor in the pandoravirus genome evolution.

This paper represents an important step forward towards a better understanding of the evolutionary processes that led to the existence of giant pandoravirus genomes, even though the deep evolution of these viruses is still shrouded in mystery.

The paper is written logically and the experimental procedures are sound.

This reviewer has only minor suggestions to improve the manuscript further:

- Virus nomenclature: Please adhere to the following guidelines: virus families are always italicized (suffix -viridae), and so are virus species names in a taxonomic context. However, neither families nor species can be isolated, manipulated etc. since they are abstract terms; hence, in these cases you would refer to 'members of the family' and write the virus names non-italicized (e.g. p. 3: "...we previously isolated Megavirus chilensis [non-italics]"). Also, host species names in virus names are not italicized, e.g. p.3: *Acanthamoeba polyphaga* mimivirus [either put all in italics when talking about the taxonomy, or no italics when talking about discoveries, manipulations etc]. Non-ICTV approved taxa should be declared as such, e.g. by adding 'proposed' in front.

- Exocytosis of pandoravirus particles: On p.4, the authors state that mature virions are released through exocytosis at 8 hours pi, but no data is shown to support this claim. Fig. S1C supposedly shows vesicle-enclosed mature virions, but a) the resolution of the images is too low to unambiguously spot a membrane, and b) how do the authors exclude that these particles are not entering the cell?

- Which reference gene set was used for CAI calculation?

- P. 6: "e.g. 90% for Mimiviridae" – does this number refer to transcribed or translated genome regions?

- What corresponds to 100% of the genes shown in Fig. S6 – all protein-coding pandoravirus genes, or only those with homologs outside pandoraviruses? This should be clarified in the legend.

- P.8 references a Fig. 6A, but there is no A and B in Fig. 6.

- P.9: The graph showing purifying selection of pandoravirus genes is shown in Fig. S11A, not D.

Reviewer #2 (Remarks to the Author):

This paper describes a number of novel (giant) pandoraviruses, with a particular focus on their comparative genomics and evolution. While the pandoviruses are indeed very interesting because of their large size, this paper unfortunately suffers from a number of very important limitations. In particular, although the analysis seems comprehensive (including the gene annotations and most of the genomics) and generally well done (and I like the RNA-Seq analysis), some of the evolutionary conclusions drawn are very bizarre and off-base.

The authors find evidence for both gene duplications and lateral gene transfer in the pandovirus genome - these conclusions seem reasonable and are supported by the data. All good. However, the trouble starts when they claim that the genes with no apparent homology to any other known genes must have been created *de novo* and then go on to explain how and why this happens. Unfortunately there some major logical flaws in this argument. Most fundamentally, just because you didn't find a homolog does not mean that it doesn't exist. What the authors are forgetting is that all homology searches are necessarily limited to those virus genomes that are present on GenBank or related databases. However, these represent a minuscule fraction of total number of viruses in nature of which we have only sampled a pathetically small number. Hence, it is perfectly possible - and I would argue likely - that these genes have homologs in currently undescribed viruses and hence have not been created *de novo* in pandoraviruses. This is hugely important and to me seems a more reasonable explanation for the data.

The evolutionary processes proposed for *de novo* gene creation are also very strange and in some instances don't make sense at all. Most telling is that during the Discussion the authors also highlight the limitations in their own thinking. All very odd. For example, I simply have no idea how genes could "randomly emerge from non-coding intergenic regions". This seems nuts to me. Apologies, but it does. The transitional steps from random to coding sequence would only be selected if these intermediate forms some functional benefit, but how can there be during the early stages when they are so unlike real genes? This entirely section of the Discussion is very, very unconvincing. Similarly, I see no evidence that pandoviruses "must exhibit some specific features leading them to favor *de novo* gene creation during their evolution". Why? Again, if the authors were to simply realise that this is all an artefact due to the tiny number of viruses sampled then all these bizarre *de novo* gene creation ideas just disappear.

More generally, the paper reads like a comparative genomics paper and hence may be better suited to a more specialist journal like GBE. It is also written in rather strange way so that it is not easy to tell whether it is a primary research or a review paper, and parts of the text are highly descriptive.

OTHER POINTS

The paper is poorly written, although some of this may reflect that the authors are not native English speakers. In particular, (i) the paper is repetitive, (ii) parts of Results are included in the Discussion, (iii) the end of the Introduction contains the Results so that it is largely a repeat of the Abstract, (iv) some of the phraseology used is loose and rather unscientific (e.g. I don't think it is right to say that the pandoviruses are most 'spectacular' viruses. Hence, the paper needs a major rewrite.

I do not understand this statement: "In general, the conceptually appealing notion (inherited from the RNA viruses) that the large proportion of ORFans found in eukaryotic dsDNA viruses would be the consequence of a large mutation rate is progressively losing ground". Please clarify.

The tree in Figure 8 is very poor (and there is no clearly description of the method actually used to infer it). While I think that gene presence/absence can show that viruses are distinct, it is a very poor way to infer phylogenetic relationships, which why the base of the tree is so star-like. I would far rather see a normal sequence-based tree utilizing the truly homologous sequences.

Reviewer #3 (Remarks to the Author):

Legendre and colleagues have isolated three new Pandoravirus from different locations worldwide. In this manuscript, they compare their genomes to the three other Pandoravirus genomes already available (two from their own laboratory) to initiate a comparative genomic analysis of the family Pandoraviridae. In particular, they use this new dataset to investigate the origin of orphan genes in the genomes of these giant viruses. This is the most interesting part of this manuscript for a general readership since it directly addresses the fascinating question of giant virus origin. Their data support the hypothesis that most of these genes are recent and originated by de novo creation.

The authors have performed a careful identification of ORFs by combining genomic, transcriptomic and proteomic approaches. This led to a dramatic reduction of the number of predicted ORFs, but, interestingly, to the detection of a high number of gene associated to non-coding transcripts. Importantly, they notice that a very high number of ORFans still remains after their stringent reannotation.

The authors then address the problem of the origin of giant viruses "gigantism". Several hypotheses have been proposed by various authors to explain this phenomenon, such as extensive HGT from their hosts, genome duplication, retention of ancestral gene from ancient cellular organisms (the hypothesis previously favoured by the authors) or else extensive de novo creation of new genes. Their analysis is carefully done. In particular, they avoid to systematically attribute the presence of cellular homologues to HGT from hosts to viruses, noticing that such homology can also testify for HGT from viruses to hosts. Although they detect a significant proportion of HGT and gene duplication, their careful analysis strongly suggests that the main factor for genome expansion in Pandoraviridae is de novo genes creation, probably by mechanisms recently identified in eukaryotes via comparative genomics.

These results are nicely discussed in some details. Notably, they could mention in addition the review by Forterre and Gaia who previously proposed this hypothesis in a recent review (*Curr Opin Microbiol.* 2016 Jun; 31: 44-49). They wrote: "it has been suggested that NCLDV genes without cellular homologues testify for an ancient cellular origin of giant viruses. However, most of these genes might have simply originated directly in viral genomes by the same mechanisms that produce new genes in cellular genomes, possibly by recruitment and extension of intergenic open reading frames (protogenes)".

It has been shown previously that protogene often encode for disordered proteins. It should be interesting to determine if the ORFans detected here are indeed enriched in disordered proteins, especially since it has been shown that: "high GC content causes orphan proteins to be intrinsically disordered" (Basile W, Sachenkova O, Light S, Elofsson A. *PLoS Comput Biol.* 2017 Mar 29; 13(3):e1005375. doi: 10.1371/journal.pcbi.1005375). Disordered proteins can interact with other host or viral proteins, modulating their activity and influencing the virus fitness. This would explain why they could be rapidly selected and fixed in a new lineage.

The authors report new data from electron microscopy suggesting that the nucleus remains close until the end of the infection process, but with morphological change (disappearance of the nucleolus). It seems that Pandoraviridae transform the nucleus itself into a viral factory. This is an exciting part of the story since several authors have discussed a possible evolutionary link between the nucleus and the viral factory of giant viruses. This could be briefly discussed.

Since Pandoraviridae encode their own transcription machinery, did the authors look for specific promoters for late genes?

The authors use several times the term "open pan genome". However, this looks like a naïve statement considering the very limited number of genomes use in this study compared to similar studies with bacterial genomes. One cannot expect a saturation curve in that case. I would thus recommend not using the term "open pan genome".

The presence of spliceosomal introns is an interesting feature of these genomes. Is this specific to Pandoraviridae or present in other families of NCLDV?

From their gene presence/absence analysis, the authors observed that Mollivirus form a clade with Pandoravirus. Is this grouping supported by previous analyses in the literature (DNA polymerase phylogeny). If correct, Molliviruses could be used as an outgroup to study Pandoravirus evolution. Surprisingly, the authors don't discuss this possibility and don't try using it in their analyses. For instance, it could be interesting to determine a core genome common and specific to Mollivirus and Pandoravirus to determine protein gain and loss after the divergence of these two lineages.

Point-by-point response

Reviewer #1 (all minor suggestions)

1) Virus nomenclature: we corrected the manuscript according to the suggestion of the reviewer, (Italics and “proposed” family names).

2) Exocytosis of pandoravirus particles: this is a fact that is now documented by a movie, added as a supplementary file. To rightfully acknowledge the author of this work, we now added the name of “Eugène Christo-Foroux” in the list of authors in the revised paper.

3) - ***Which reference gene set was used for CAI calculation?***

The Reference codon usage was calculated on the most expressed protein coding genes in *A. castellanii*. Their list is now given in Supplementary dataset S3.

4) - ***P. 6: "e.g. 90% for Mimiviridae" – does this number refer to transcribed or translated genome regions?***

90% correspond to the translated genome, as it is now indicated.

5) - ***What corresponds to 100% of the genes shown in Fig. S6 – all protein-coding pandoravirus genes, or only those with homologs outside pandoraviruses? This should be clarified in the legend.***

Fig. S6 concerns all the Pandoravirus proteins with a significant match outside of the *Pandoraviridae*, as now noted in the legend.

6) - ***P.8 references a Fig. 6A, but there is no A and B in Fig. 6.***

A and B are now added to the figure.

7) - ***P.9: The graph showing purifying selection of pandoravirus genes is shown in Fig. S11A, not D.***

This is now corrected.

Reviewer #3 (all minor suggestions)

1) As suggested by the reviewer, the recent review by Forterre and Gaia (Curr Opin Microbiol. 2016) is now cited (end of page 12)

2) ***"It should be interesting to determine if the ORFans detected here are indeed enriched in disordered proteins"***.

After specifically investigating this possibility using the Disopred software, we found that ORFans are indeed slightly more disordered than non-ORFans. This is now acknowledged with the following sentence:

"It has also been suggested that proteins encoded by *de novo* created genes might be enriched in disordered regions⁴². Accordingly we observed a slightly albeit significant ($p < 10^{-15}$, Wilcoxon signed-rank test) higher fraction of predicted disordered residues⁴³ in ORFans (14%) versus non ORFans (11%)."

3) ***A possible evolutionary link between the nucleus and the viral factory of giant viruses could be briefly discussed.***

We choose not to follow this suggestion. The pandoraviruses are not unique among large DNA viruses in using the cell nucleus as their "viral factory" and our work does not contribute any new data in favor or against the viral eukaryogenesis hypothesis. In addition, reviewer #2 would probably not be in favor of our revised manuscript devoting more space to additional evolutionary speculations.

4) ***Since Pandoraviridae encode their own transcription machinery, did the authors look for specific promoters for late genes?***

The comprehensive analysis of the transcription time course is currently ongoing. Right now it appears that there is no clear signal for late genes based on the proteome composition of the pandoravirions. We believe that a detailed comparative study of the transcriptome of the various pandoravirus species is outside of the scope of this (already data-loaded) paper.

5) ***The authors use several times the term "open pan genome". One cannot expect a saturation curve in that case. I would thus recommend not using the term "open pan genome"***.

We agree with the reviewer that more species would be better. However, our statistical analysis already indicates that the shape of the saturation curve is that expected from an open pan genome (Statistics in Fig. 2B, the alpha parameter value of the Heap's law fitted to the curve is less than one, see reference 53). Symmetrically, the "core" saturation curve suggests that 6 isolates are enough to converge to a stable estimate of the core gene set. This is now better explained in the legend of Fig.2. Nevertheless, we concluded the corresponding section in a less peremptory manner (page 6).

6) ***The presence of spliceosomal introns is an interesting feature of these genomes. Is this specific to Pandoraviridae or present in other families of NCLDV?***

We now added in the text (page 5):

"Although spliceosomal introns are found in other viruses with a nuclear phase such as the chloroviruses¹¹, pandoraviruses are the only ones for which spliceosomal introns have been validated for more than 10% of their genes". The corresponding reference has been added to the reference list.

7) From their gene presence/absence analysis, the authors observed that Mollivirus forms a clade with Pandoravirus. Is this grouping supported by previous analyses in the literature (DNA polymerase phylogeny).

Yes, this affinity was already noticed in the original description of Mollivirus. This is now indicated in the text by the original reference and shown again in Fig. S14.

8) Molliviruses could be used as an outgroup to study Pandoravirus evolution.

We did not attempt this exercise for two main reasons. We believe that more Mollivirus-like isolates are needed to delineate the core gene content of an eventual *Molliviridae* family. Second, the marginal phylogenetic affinity of Mollivirus with the pandoraviruses only involves 46 genes specifically common to Mollivirus and pandoraviruses, i.e. only 10% of the core gene set (455 genes) of the *bona fide* pandoraviruses (Fig. 2.) We do not believe that it is reasonable to base an analysis of the pandoravirus genome evolution on such a small subset of genes.

Reviewer #2

As written in its report, this reviewer generally found the manuscript interesting and the genome reannotation and comparative analyses well done. His criticisms almost entirely focus on the interpretation of our data, deemed as “containing major logical flaws”. The reviewer is opposing two main arguments to our interpretation of the large proportion of ORFans as possibly resulting from *de novo* gene creation:

- 1) these ORFans are simply a consequence of our very limited knowledge of the total virosphere.
- 2) the hypothesis of *de novo* gene creation (i.e. functional genes emerging from non-coding sequences, then becoming selected) “does not make sense at all”.

Our response to the first point is the following:

We do agree with this reviewer that only a minuscule fraction of viruses in nature has been sequenced. However, such an argument only provides an explanation to the large proportion of ORFans, but not to their distinctive statistical properties (shown in Fig. S13 B, C, D) and their statistical resemblance to the pandoravirus intergenic regions. Moreover, the increasing resemblance of the core, clade-specific, and strain-specific ORFans with the intergenic regions definitely suggests an evolutionary model according to which *de novo* created genes are progressively becoming undistinguishable from more ancestral genes. Finally, the increasing proportion of ORFans in core, clade-specific, and strain-specific (Fig. S11 D) validated genes suggests an historical process rather than the mere absence of homologs due to the small fraction of sequenced viruses. What our data shows is that ORFans are not just “regular” virus genes with no known homologs, but are enriched in qualitatively different genes, that can be possibly *de novo* created.

This point is now clearly made in the discussion with the following sentences (page 12):

“Finally, it would be tempting to explain the large proportion of ORFans by our minuscule knowledge of the global virosphere, hence by the absence of homologs in the databases. However, their differential distribution among core, clade-specific and strain-specific gene sets (Fig. S11 D), as well as their distinctive statistical properties (Fig. S13) suggest that ORFans are not just regular genes missing from the database.”

The second main point raised by reviewer #2 (i.e. the implausibility of *de novo* gene creation) was already addressed in the penultimate paragraph of the discussion.

There we present the inherent main problem with the “*de novo* gene creation” hypothesis, namely the difficulty to explain how newly created genes could be retained long enough to acquire a selectable impact on the virus fitness.

Unfortunately, the reviewer interpreted our cautious and objective discussion as “highlighting the limitations in our own thinking”.

To this, we oppose two arguments:

- 1) we are not proposing a brand new model, but simply referring to a model previously proposed and published by noted authors (e.g. Carvunis et al., Nature 2012, Pubmed: 22722833, Wilson et al., Nat. Ecol. Evol., 2017, Pubmed: 28642936, and encouraged to do more so by reviewer #3). Thus, to at least a fraction of contemporary researchers, such a model is a rational possibility,

2) Finally, despite our current difficulty to explain how new functions could emerge and be selected, it is self-evident that “*de novo*” gene creation had to happen repetitively during evolution for Life itself to exist.

Minor points

1) **“The paper is poorly written ... major rewrite”.**

Although the other reviewers did not complain about the style of the manuscript, we had it read by native English speakers who suggested a few changes that have been made.

2) **“Part of results are included in the discussion”.**

The results included in the discussion correspond to the additional analyses we performed to establish the possibility of the *de novo* gene creation. Presenting these analyses out of this context (that has to be presented first) would be illogical. The two other reviewers did not criticize the organization of the manuscript.

3) **“the end of the introduction contains the results ...”**

The overlap between the abstract and the introduction should not be a problem according to the journal guideline to authors.

4) **“It is not right to say that pandoraviruses are spectacular”**

The word “spectacular” was changed into “complex”

5) **“I do not understand this statement”**

the statement “in general losing ground”, was deleted.

6) The delineation of (distant) virus families based on the clustering of gene contents (presence/absence of genes) highlights the differences in physiology of the virus families. In contrast, alignment-based phylogenies reconstruct the common ancestry of the small number of proteins common to all DNA virus families. The corresponding signals might be blurred by lateral exchanges. As mentioned by this reviewer, the purpose of Figure 8 is to illustrate to what extent the various virus families are distinct. However, the star-like structure of the tree persists when using a single conserved gene such as the DNA polymerase (now added as supplementary figure S14, as also requested by reviewer #3). The quality of the tree is thus not in question, but is consistent with an early divergence of the virus families from their (hypothetical) common ancestor (if any). A detailed description of the procedures used in building the trees shown in Fig. 8 and the new Fig. S14 is given in the supplementary materials and methods section.

REVIEWERS' COMMENTS:

Please note that although Reviewer #1 doesn't have Remarks to the Author, in his/her Remarks to the Editor, this reviewer feels the authors have addressed all concerns raised by the reviewers.

Reviewer #2 (Remarks to the Author):

This paper is definitely an improvement over the original and I have no major objections to its publication. The writing is a lot better. That said, I still disagree with the authors on the two fundamental points I raised in my last review and which I don't think they have done a particularly good job at addressing. First, to me, it is blindingly obvious that some of the ORFans will not turn out to be so once we sample more of the virosphere. Again, we have sampled almost nothing (statistically) of the virosphere and to think that our notion of ancestry is going to be unchanged as we sample more seems hopelessly naïve. More directly, the authors argument based on differential distribution (lines 470-473) really makes no sense to me. Similarly, I remain deeply unconvinced by the de novo gene creation from non-coding DNA idea and the authors have not presented a single piece of strongly supportive evidence for it. As I said in my last review, what I just don't understand how the intermediate steps between non-coding DNA and functional gene are selectively favored. An explanation would be nice. However, as long as this is presented as an hypothesis rather than a fact I can live with it.

Reviewer #3 (Remarks to the Author):

In my opinion, the authors have answered and take into account correctly most of the reviewers comments and they provide reasonable arguments to refute or decline some of them. In particular, I completely agree with their rebuttal of reviewer #2 criticisms concerning the problem of de novo gene creation. They added in the revised manuscript a sentence that will certainly help clarifying this point.

Reviewer #1:

No further remarks to the authors.

In his/her Remarks to the Editor, this reviewer feels the authors have addressed all concerns raised by the reviewers.

Reviewer #3:

In my opinion, the authors have answered and take into account correctly most of the reviewers comments and they provide reasonable arguments to refute or decline some of them. In particular, I completely agree with their rebuttal of reviewer #2 criticisms concerning the problem of de novo gene creation. They added in the revised manuscript a sentence that will certainly help clarifying this point.

According to the opinion of these two referees, the manuscript is now satisfactory as it is. To be fair to their opinions and maintain the originality of the paper, we had to keep additional changes to a minimum (and within the imposed word count limit). Within these constraints we attempted to further accommodate the suggestion of reviewer's 2, although reviewer's #3 feels that we adequately rebutted his arguments against de novo creation (see above).

Reviewer #2:

The writing is a lot better. That said, I still disagree with the authors on the two fundamental points I raised in my last review and which I don't think they have done a particularly good job at addressing. First, to me, it is blindingly obvious that some of the ORFans will not turn out to be so once we sample more of the virosphere.

Again, we have sampled almost nothing (statistically) of the virosphere and to think that our notion of ancestry is going to be unchanged as we sample more seems hopelessly naïve.

More directly, the authors' argument based on differential distribution (lines 470-473) really makes no sense to me. Similarly, I remain deeply unconvinced by the de novo gene creation from non-coding DNA idea and the authors have not presented a single piece of strongly supportive evidence for it. As I said in my last review, what I just don't understand how the intermediate steps between non-coding DNA and functional gene are selectively favored. An explanation would be nice. However, as long as this is presented as an hypothesis rather than a fact I can live with it.

Fortunately, Reviewer #2 will now find acceptable the new version (e.g. "could live with it") if we present the *de novo* gene creation as an hypothesis rather than a fact. Although this was already done in the first revised version of the manuscript, we emphasized this even further in the final version. The "hypothetical" nature of the model is now cited 6 times:

-Page 12

Line 15, line 21

-Page 13

Line 18, Line 23, line 29, line 47

Our final misunderstanding remains about the interpretation of the numerous ORFans (but in fact Pandoravirus-specific genes) characterizing the Pandoraviridae. Although the other reviewers felt we convincingly explained this point, Reviewer #2 still does not get it.

Here is our rationale again, exposed somewhat differently:

We of course agree with this reviewer that it is “obvious that some of the ORFans will not turn out to be so once we sample more of the virosphere”. However, what characterizes the Pandoraviridae is the unprecedented number of genes that they have in common that are NOT found in other virus families (“Pandoraviridae-specific ORFans). We are not referring to genes that have no homologs at all, but to genes that have homologs within the Pandoraviruses, but none outside this family. This is what suggests that these genes appeared (recently?) within this family, and did not derived from ancestral genes already present in the ancestor of all DNA viruses (provided it existed, but this is a different story). So, if the number of *bona fide* ORFan gene will decrease with time, the number of these genes uniquely found in Pandoraviridae (Pandoraviridae-specific ORFans) might not decrease much, except through eventual horizontal transfers between virus of different families.

To try to clarify this point, once again, we replaced the previous paragraph by the following:
Page 11, Line 36-43:

The proportion of ORFans (i.e. proteins without homologs in the databases) obviously depends on our limited knowledge of the virosphere. However, what characterizes the Pandoraviridae is the unprecedented number of family-specific ORFans they share, the increase of their proportion among core, clade-specific and strain-specific gene subsets (Supplementary Fig. 11 D), as well as their distinctive statistical properties (Supplementary Fig. 13). Altogether, this suggests that the Pandoravirus-specific ORFans are not just ancestral genes missing from the database, but genes with histories confined within the Pandoraviridae.

We hope that this will clarify the issue. The misunderstanding probably originated from the confusion between “ORFans” (no homologs at all), and “family-specific ORFans” (only present in Pandoraviruses).

Finally, reviewer #2 persisted in complaining (albeit nicely) about the lack of explanation about the intermediate steps allowing emerging proteins to become selected.

We made clear, in the previous version of the manuscript that this was a major difficulty, although not only for us, but for all the other researchers that have previously proposed the *de novo* gene creation hypothesis in the context of other organisms. As others, we do not have yet an explanation, but we preferred to be honest about it, instead of hand waving. However, it is also clear that no alternative model can provide a parsimonious explanation for the emergence of the million of genes (ancestral or not) that are presently unique to the

virosphere. It is also clear that a straightforward Darwinian “selection of the fittest” model, very rarely apply to extreme parasites that enjoy a “free meal” situation, and population bottlenecks allowing the evolutionary success of suboptimal variants. In such condition, the burden of translating a protein initially emerging without function might be tolerated for quite a while.

Gracefully, the reviewer is now accepting this sad situation (*“An explanation would be nice. However, as long as this is presented as an hypothesis rather than a fact I can live with it”*) provided we insist on the hypothetical nature of the model, as we did (see above).

We hope that you will find this final version of the manuscript fully acceptable for publication.

Best regards,

M. Legendre, C. Abergel, J.-M. Claverie